# The Influence of Treatment with Ledipasvir/Sofosbuvir on Growth Parameters in Children and Adolescents with Chronic Hepatitis C

**DOI:** 10.3390/v14030474

**Published:** 2022-02-25

**Authors:** Maria Pokorska-Śpiewak, Anna Dobrzeniecka, Magdalena Marczyńska

**Affiliations:** 1Department of Children’s Infectious Diseases, Medical University of Warsaw, 01-201 Warsaw, Poland; magdalena.marczynska@wum.edu.pl; 2Department of Pediatric Infectious Diseases, Regional Hospital of Infectious Diseases in Warsaw, 01-201 Warsaw, Poland; adobrzeniecka@zakazny.pl

**Keywords:** body mass index, children, chronic hepatitis C, growth, ledipasvir/sofosbuvir

## Abstract

Background: There are limited data available on the influence of direct-acting antivirals used to treat chronic hepatitis C (CHC) on growth in children. In this study, we aimed to analyze the growth parameters in children treated with ledipasvir/sofosbuvir (LDV/SOF). Methods: We included 38 patients (16 girls and 22 boys) aged 10–17 years treated with LDV/SOF for CHC (33 infected with genotype 1 and 5 with genotype 4; 36 were treated for 12 weeks, and 2 for 24 weeks according to the current guidelines). Patient weight and height were measured at baseline, after 4 weeks of treatment, at the end of the treatment (EOT), and 12 weeks and one year after the EOT. Body mass index (BMI), BMI z and height-for-age (HA) z scores were calculated according to the WHO Child Growth Standards and Growth reference data using the WHO anthropometric calculator AnthroPlus v. 1.0.4. In addition, correlations between BMI z scores and liver fibrosis (liver stiffness measurement, LSM), the aspartate transaminase (AST)-to-platelet ratio index (APRI), fibrosis-4 index (FIB-4) and liver steatosis (controlled attenuation parameter, CAP) were analyzed. Results: At baseline, 5/38 (13%) patients were obese (BMI z score > 2 SD), 4/38 (11%) were overweight, and 29 (76%) were normal. A significant increase was observed in mean weight, height and BMI both 12 weeks and one year after the treatment compared to the baseline, whereas no differences were observed for BMI z scores and HA z scores. Baseline BMI z scores correlated with alanine aminotransferase levels (r = 0.33, 95% CI 0.01–0.58, *p* = 0.04), LSM (r = 0.40, 95% CI 0.09–0.65, *p* = 0.01), the APRI (r = 0.33, 95% CI 0.02–0.59, *p* = 0.03), and the CAP (r = 0.40, 95% CI 0.08–0.64, *p* = 0.01). No similar correlations were reported at 12 weeks posttreatment. Conclusions: Treatment with LDV/SOF in children with CHC (genotypes 1 and 4) did not negatively influence the patients’ growth. However, higher baseline BMI z scores correlated with more advanced liver fibrosis and steatosis in children with CHC.

## 1. Background

Therapies based on direct acting antivirals (DAAs) are currently considered a standard of care in patients with chronic hepatitis C (CHC). In 2017, the first DAA, ledipasvir/sofosbuvir (LDV/SOF), was approved for pediatric patients infected with hepatitis C virus (HCV) [1]. Since 2020, LDV/SOF may be used in patients as young as 3 years of age [2,3]. The high efficacy and excellent safety profile of LDV/SOF were confirmed in both clinical trials and real-life studies [4,5,6]. Interferon (IFN)-based therapies, which were considered a standard of care for CHC before the era of DAAs, had a significant impact on the growth parameters in pediatric patients [7]. Treatment with pegylated IFN was associated with significant changes in body weight, growth, and body mass index (BMI) in children [7]. These effects were generally reversible; however, high-for-age z scores (HA z scores) did not return to baseline after two years of observation, which indicates the long-term influence of this therapy on the children’s growth. As the history of DAA use in children is relatively short, data on their influence on growth parameters are scarce, and no longitudinal observations exist. Thus, in this study, we aimed to analyze the growth parameters in children infected with genotypes 1 and 4 HCV during and after treatment with LDV/SOF. In addition, correlations between BMI and liver inflammation, fibrosis, and steatosis were analyzed.

## 2. Materials and Methods

### 2.1. Study Group

All consecutive patients aged 10–17 years infected with genotypes 1 and 4 HCV and treated with LDV/SOF between August 2019 and August 2021 in the real-life therapeutic program ‘Treatment of Polish Adolescents with Chronic Hepatitis C Using Direct Acting Antivirals (POLAC PROJECT)’ were included in this study. The efficacy and side effects of LDV/SOF treatment in participants aged 12–17 years included in this project were described in detail in our previous study [6]. CHC was diagnosed in patients with over a 6-month duration of HCV infection confirmed with positive HCV RNA testing using a quantitative real-time polymerase chain reaction (RT–PCR, Abbott RealTime HCV, Abbott Laboratories, Abbott Park, IL, USA; measurement linearity range 12–1.0 × 10^8^ IU/mL). Children qualified for participation in this program irrespective of previous ineffective treatment with IFN or the extent of liver fibrosis. Duration of the treatment was 12 weeks; however, cirrhotic patients infected with HCV genotype 1 and with a history of previous ineffective IFN-based treatment qualified for a 24-week therapy [8]. All participants were followed every 4 weeks during the treatment, at the end of the therapy (EOT), and at week 12 posttreatment to assess sustained virologic response (SVR12), which indicated the efficacy of the treatment. An additional visit at one year (52 weeks) after the EOT was scheduled in cases with significant fibrosis at baseline and for volunteer patients without liver fibrosis.

The BMI, BMI standard deviation (SD) (BMI z scores) and high-for-age z scores (HA z scores) were calculated using the WHO anthropometric calculator AnthroPlus v.1.0.4. according to the WHO Child Growth Standards and Growth reference data. Obesity was diagnosed in participants with a BMI z score > 2 SD and overweight > 1 SD.

### 2.2. Evaluation of Liver Fibrosis and Steatosis

Noninvasive methods, including transient elastography (TE) and serum biomarker analysis, were used to evaluate liver fibrosis and steatosis.

TE was performed using the FibroScan device (Echosens, Paris, France). We simultaneously obtained liver stiffness measurement (LSM) and controlled attenuation parameter (CAP). The adequacy of the measurement was assessed by the FibroScan device. The examination was considered successful when 10 valid measurements were obtained with at least a 60% success rate and an interquartile range (IQR) not exceeding 30% of the median LSM value [9]. The final LSM result was assessed in kilopascals (kPa) and it was expressed as the median value of at least 10 valid measurements. It corresponded to liver fibrosis on the METAVIR scale according to the Castera TE cutoffs [10]. Liver fibrosis was considered significant when the LSM median was >7 kPa, corresponding to a METAVIR F score ≥ 2 points. Cirrhosis (F4) was diagnosed when LSM was ≥12.5 kPa. The CAP values ranged between 100 and 400 decibels per meter (dB/m), and CAP > 238 dB/m corresponded with steatosis [11,12]. TE examination was performed simultaneously with the biomarker evaluation on the day the patient started treatment, at week 12 posttreatment, and one year after the end of the treatment.

Commercially available laboratory kits were used for biochemical serum testing. For both alanine and aspartate aminotransferase (ALT and AST) serum levels, 40 IU/L was considered the upper limit of normal (ULN). We calculated two indirect fibrosis biomarkers, namely, the aspartate transaminase-to-platelet ratio index (APRI) and the fibrosis-4 index (FIB-4), according to the published analytic recommendations [13,14]:

APRI = [(AST (IU/L)/AST ULN)/Platelet count (10^9^/L)] × 100

$$\text{FIB-4}=[\mathrm{Age}\;(\mathrm{years})\times \mathrm{AST}\;(\mathrm{IU}/L)]/[\mathrm{Platelet}\;\mathrm{count}\;(10^{9}/L)\times \sqrt{\mathrm{ALT}\;\left(\mathrm{IU}/L\right)}]$$

The following cutoffs for these biomarkers were considered: APRI > 0.5 and FIB-4 > 1.45, which suggests significant fibrosis, and APRI > 1.5, which suggests cirrhosis [13,15].

### 2.3. Statistical Analysis

Categorical variables were expressed as numbers and percentages of the total, and they were compared using either the chi-square test or Fisher’s exact test, as appropriate. Continuous variables were tested for normal distribution using the Kolmogorov–Smirnov test. They were expressed as the mean +/− SD or the median with interquartile range (IQR), as appropriate, and they were compared using the Student’s t-test or the Mann–Whitney test. To compare continuous variables between more than two groups, repeated measures analysis of variance (ANOVA) testing was performed. Only subjects with complete data were included in these analyses. Pearson coefficients, r (95% confidence interval, CI), were calculated to analyze the correlation between the continuous variables. All statistical analyses were performed using MedCalc Statistical Software version 20.009 (MedCalc, Ostend, Belgium). A two-sided *p* value of <0.05 was considered significant.

### 2.4. Ethical Statement

We collected written informed consent from all the patients and/or their parents/guardians before their inclusion in the study. The local ethics committee of the Medical University of Warsaw approved this study (approval number: KB/87/2019; date of approval: 13 May 2019). The investigation was performed in accordance with the ethical standards in the 1964 Declaration of Helsinki and its later amendments.

## 3. Results

### 3.1. Study Group

We included 38 patients (16 girls and 22 boys) aged 10–17 years treated with LDV/SOF (33 infected with genotype 1 and the remaining 5 with genotype 4). According to the current guidelines, thirty-six participants were treated for 12 weeks, and two were treated for 24 weeks. At baseline, 5/38 (13%) patients were obese (BMI z score > 2 SD), 4/38 (11%) were overweight, and 29 (76%) were normal. The baseline characteristics of the study group according to the initial BMI z scores are presented in Table 1. All participants achieved SVR12 except one obese patient with a baseline BMI z score of 3.28 who was lost to follow-up after 4 weeks of treatment (at this point, his HCV RNA was undetectable, and he completed the 24-week therapy). Two other patients (one normal and one overweight) missed the EOT visit due to the COVID-19 pandemic. One-year follow-up examinations were performed in three patients with baseline liver fibrosis > F1 and in five patients without liver fibrosis. Among them, three patients were normal weight, four were overweight, and one was obese.

### 3.2. Growth Parameters during and after Treatment

Growth parameters during and after treatment with LDV/SOF in patients up to 12 weeks after EOT are presented in Table 2. In this analysis, we included 35 patients for whom complete data were available. A significant increase in patients’ weight, height, and BMI values was observed during and after 12 weeks of treatment (*p* < 0.001; *p* < 0.001; and *p* = 0.006, respectively), whereas no significant difference was observed for the BMI z scores and HA z scores (Table 2). In eight patients with a one-year follow-up, a significant increase in weight, height, and BMI occurred compared to the baseline (*p* = 0.008; *p* = 0.04; and *p* = 0.02), whereas the BMI z scores and HA z scores were similar (Table 3).

In addition, an analysis was performed comparing the mean increase in BMI values (at 12 weeks and at one year after EOT compared to the baseline) between the study group and a control group consisting of age- and sex-matched controls, using the WHO Growth reference data. The mean increase in BMI values in our study group from baseline to 12 weeks posttreatment was 0.97 (95% CI 0.26–1.68), and from baseline to one year posttreatment it was 2.86 (95% CI 0.37–5.35). The relevant increases in BMI values in the control group were 0.27 (95% CI 0.26–0.29) and 0.69 (95% CI 0.56–0.81), respectively. The differences between the study and control groups were insignificant; however, a trend towards higher BMI increases in the study group was observed (*p* = 0.05 for the increase at 12 weeks posttreatment and *p* = 0.06 for the increase at one year posttreatment, respectively).

The proportions of children with different BMI z scores during and after treatment did not show any significant differences (Figure 1).

### 3.3. Correlations between BMI and Liver Inflammation, Fibrosis, and Steatosis

At baseline, overweight and obese children presented with significant liver fibrosis, including cirrhosis and steatosis, more often than children with normal BMI z scores (*p* = 0.004 and *p* = 0.007, respectively) (Table 1). Baseline BMI z scores correlated positively with ALT levels (r = 0.33, 95% CI 0.01–0.58, *p* = 0.04), LSM (r = 0.40, 95% CI 0.09–0.65, *p* = 0.01), the APRI (r = 0.33, 95% CI 0.02–0.59, *p* = 0.03), and the CAP (r = 0.40, 95% CI 0.08–0.64, *p* = 0.01). No similar correlations were reported at 12 weeks posttreatment. Analysis performed one year after the end of treatment revealed a correlation between BMI z scores and ALT levels. Still, no correlation with liver fibrosis or steatosis was observed at this point (Table 4).

## 4. Discussion

DAA-based therapies for CHC are highly effective and well-tolerated. However, their short- and long-term impact on the growth of pediatric patients requires evaluation. Our cohort revealed that therapy with LDV/SOF was not associated with any BMI z score or HA z score decrements at 12 weeks and one year after the treatment compared to the baseline. A significant increase in patients’ weight, height, and BMI was observed during both periods. Our results are similar to the observations of Serranti et al., who treated 78 participants aged 12 to <18 years with LDV/SOF and confirmed the significant increase in weight and height from baseline to 12 weeks from the end of the therapy (mean values from 58 to 60.2 kg, *p* < 0.0001, and from 162.7 to 164 cm, *p* < 0.0001, respectively); however, no significant difference was found in BMI, BMI z, or HA z scores [5]. In a single Egyptian center study by Fouad et al., which investigated the effectiveness of LDV/SOF in 51 adolescents aged 11–17.5 years infected with HCV genotype 4, a significant increase in individual weight, height and BMI values was observed at the 6-month follow-up compared to the baseline (49.3–53.5 kg, *p* < 0.01; 158.2–159 cm, *p* < 0.01; 19.5–21 kg/cm^2^, *p* < 0.01) [16]. In another Egyptian study, a BMI difference was observed between the baseline and patients 12 weeks after treatment, with a significant improvement reported (from 19.6 to 21.2, *p* < 0.001) [17]. To the best of our knowledge, our study provides the most extended observation period on the influence of DAA treatment on growth parameters in children, documenting no longer-term impact on growth progression in pediatric patients. The expected increase in BMI values in our children treated with LDV/SOF was achieved up to one year after the treatment. Compared to IFN-based therapies, which led to an even 2-year decrease in HA z scores [7], our observations confirmed another advantage of DAA treatment in children.

The BMI z score did not influence the response to LDV/SOF treatment in our cohort, as the per-protocol analysis revealed 100% SVR12 in the whole group [6]. Similar observations were made by other authors treating pediatric patients [16]. This is in contrast with the influence of BMI on the effectiveness of IFN-based therapies, as it was shown that an increase of one z score unit in the baseline BMI z scores was associated with a 12% decrease in the probability of SVR in children treated with IFN plus ribavirin [18]. In addition, there is some evidence in adult patients treated with DAAs that high BMI values may be a negative predictor for achieving SVR. Thus, an expanded treatment schedule in patients with high BMI values should be considered [19]. Our study found that higher baseline BMI z scores correlated positively with liver inflammation, fibrosis, and steatosis. This observation confirms previous reports on the positive association between liver fibrosis and steatosis and BMI z scores in children with CHC [18,20,21]. Liver steatosis is usually associated with genotype 3 HCV [20]. In this study, we did not include children infected with this genotype, as treatment with LDV/SOF is not dedicated to this group of patients. However, we revealed that steatosis is also common in children infected with genotypes 1 and 4 and is probably related to metabolic factors, as 4/5 patients with steatosis in our group presented with BMI z scores > 1. In addition, 33% of the patients with BMI z scores over 1 SD were cirrhotic. Thus, overweight and obese children with CHC are at risk of much more progressive liver disease and should be promptly qualified for treatment with DAAs.

The main limitation of our study is the relatively small number of included patients. However, all consecutive patients aged 10–17 years infected with HCV genotypes 1 and 4 that were referred to our department were included. Studies on the large groups of pediatric patients in this area are unavailable. To the best of our knowledge, we have presented the second report on real-life experiences with LDV/SOF in adolescents from Europe, after the Italian cohort [5]. In particular, the group of patients with longer-term observation is limited. However, it represents the most extended available observation period on growth in pediatric patients treated with LDV/SOF, which makes our observations unique. Second, we should mention gaps in the available data that result from the significant disruption caused by the COVID-19 pandemic.

In conclusion, treatment with LDV/SOF in children with CHC (genotypes 1 and 4) did not show any short- or longer-term negative influence on pediatric patients’ growth. Higher baseline BMI z scores correlated with more advanced liver fibrosis and steatosis. Thus, overweight and obese patients should be qualified for treatment as soon as possible to prevent the significant progression of liver disease in these groups.

## Figures and Tables

**Figure 1 viruses-14-00474-f001:**
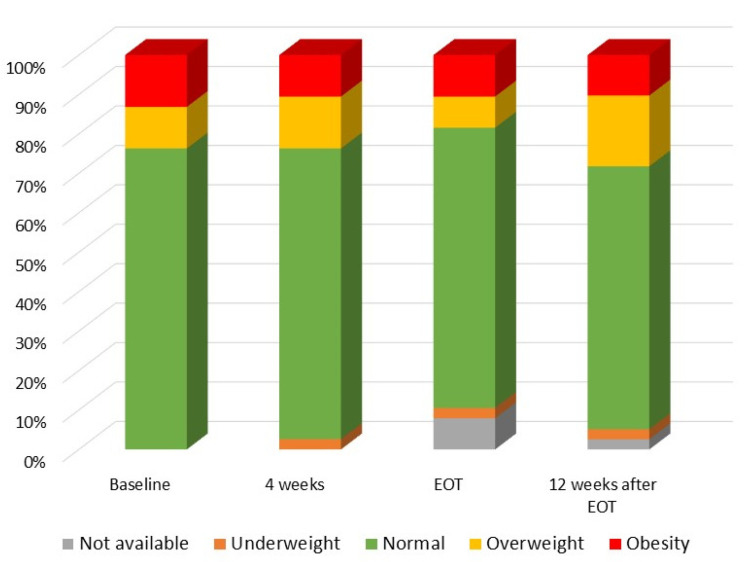
Proportions of children with different BMI z scores. Obesity was defined as BMI z score > 2 standard deviations (SD); overweight as BMI z score > 1 to 2 SD; normal as BMI z score between −2 to 1 SD; underweight <−2 SD. One patient with a baseline BMI z score of 3.28 was lost to follow-up after week 4 of treatment; two patients (one normal and one overweight) missed the EOT visit due to the COVID-19 pandemic. EOT—end of treatment.

**Table 1 viruses-14-00474-t001:** Baseline characteristics of 38 patients treated with ledipasvir/sofosbuvir according to the baseline body mass index z scores (BMI z scores).

Characteristics		All Patients (*n* = 38)	Patients with BMI z Score ≥ 1 SD (*n* = 9)	Patients with BMI z Score < 1 SD (*n* = 29)	*p* Value
Sex	Male	16 (42)	4 (44)	12 (41)	0.87
	Female	22 (58)	5 (56)	17 (59)	
Age	Mean ± SD	14 ± 2.0	14.5 ± 2.0	14 ± 1.85	0.48
HCV genotype	1	33 (87)	8 (89)	25 (86)	1.0
	4	5 (13)	1 (11)	4 (14)	
Mode of infection	Mother-to-child transmission	31 (82)	6 (67)	25 (86)	0.32
	Unknown	7 (18)	3 (33)	4 (14)	
Previous ineffective treatment with interferon plus ribavirin		15 (39)	5 (56)	10 (34)	0.26
ALT	IU/mL, Median (IQR)	37 (30; 52)	40 (32; 60)	36 (29; 45)	0.36
AST	IU/mL, Median (IQR)	37 (32; 49)	48 (33; 55)	36 (32; 47)	0.31
HCV viral load	IU/mL, Median (IQR)	5.36 × 10^5^ (1.76 × 10^5^; 1.06 × 10^6^)	6.89 × 10^5^ (1.8 × 10^5^; 1.9 × 10^6^)	4.89 × 10^5^ (1.14 × 10^5^; 1.06 × 10^6^)	0.64
SVR12	Undetectable HCV RNA at 12 weeks after EOT	37 (97) *	8 (89) *	29 (100)	0.07
Liver fibrosis (LSM corresponding to METAVIR scale)	F0/F1	33 (87)	6 (67)	27 (93)	0.004
	F2	2 (5)	0	2 (7)	
	F4	3 (8)	3 (33)	0	
LSM	kPa, Median (IQR)	5.2 (4.3; 6.3)	5.2 (4.3; 12.7)	5.2 (4.3; 5.9)	0.39
CAP	>238 dB/m (steatosis)	5 (13)	4 (44)	1 (3)	0.007
CAP	dB/m, Median (IQR)	192 (178; 213)	237 (213; 248)	191 (173; 202)	0.0009
APRI	Median (IQR)	0.32 (0.26; 0.42)	0.28 (0.23; 0.38)	0.32 (0.27; 0.38)	0.90
FIB-4	Median (IQR)	0.34 (0.28; 0.39)	0.34 (0.27; 0.50)	0.33 (0.29; 0.39)	0.82

Data are presented as the number (%) or median (IQR), as appropriate. * One obese patient was lost to follow-up after 4 weeks of treatment (at this point, his HCV RNA was undetectable, and he completed the 24-week therapy). ALT—alanine aminotransferase; APRI—aspartate transaminase-to-platelet ratio index; AST—aspartate aminotransferase; BMI—body mass index; CAP—controlled attenuation parameter; FIB-4—fibrosis-4 index; LSM—liver stiffness measurement; SD—standard deviation; SVR—sustained virologic response.

**Table 2 viruses-14-00474-t002:** Growth parameters during and after treatment with ledipasvir/sofosbuvir in 35 patients up to 12 weeks after EOT.

Growth Parameter	Baseline	After 4 Weeks of Treatment	EOT	12 Weeks after EOT	*p*
Weight [kg]	56.9 ± 13.9	57.4 ± 13.8	57.1 ± 14.7	61.8 ± 15.3	<0.001
Height [cm]	164.5 ± 9.8	165.1 ± 9.7	165.7 ± 9.6	167.6 ± 9.3	<0.001
Height-for-age z score	0.20 ± 1.2	0.28 ± 0.18	0.27 ± 1.14	0.29 ± 1.03	0.32
BMI	20.8 ± 4.1	20.6 ± 3.9	20.6 ± 4.2	21.6 ± 4.2	0.006
BMI z score	0.17 ± 1.18	0.08 ± 1.21	0.04 ± 1.29	0.22 ± 1.32	0.07

Data are presented as the means ± standard deviations. Only patients with complete data available were included in this analysis. BMI—body mass index; EOT—end of treatment.

**Table 3 viruses-14-00474-t003:** Growth parameters at baseline and one year after treatment with ledipasvir/sofosbuvir in 8 patients.

Growth Parameter	Baseline	One Year after EOT	*p*
Weight [kg]	62.5 ± 16.5	75.6 ± 19.9	0.008
Height [cm]	171.6 ± 13.3	176.3 ± 10.4	0.04
High-for-age z score	0.97 ± 0.95	1.01 ± 1.60	0.82
BMI	21.6 ± 4.2	24.4 ± 5.8	0.02
BMI z score	0.42 ± 1.13	0.79 ± 1.50	0.26

Data are presented as the means ± standard deviations. Only patients with complete data available were included in this analysis. BMI—body mass index; EOT—end of treatment.

**Table 4 viruses-14-00474-t004:** Correlation between body mass index z scores and liver inflammation, fibrosis and steatosis before and after treatment with ledipasvir/sofosbuvir.

Feature	Baseline	12 Weeks after EOT	One Year after EOT
Number of patients	38	37 *	8 **
ALT (IU/mL)	0.33 (0.01; 0.58), *p* = 0.04	0.13 (−0.19; 0.44), *p* = 0.41	0.81 (0.25; 0.96), *p* = 0.01
AST (IU/mL)	0.29 (−0.02; 0.56), *p* = 0.06	−0.17 (−0.47; 0.16), *p* = 0.31	−0.16 (−0.77; 0.61), *p* = 0.69
LSM	0.40 (0.9; 0.65), *p* = 0.01	0.06 (−0.67; 0.73), *p* = 0.88	0.27 (−0.53; 0.82), *p* = 0.5
CAP	0.40 (0.08; 0.64), *p* = 0.01	0.23 (−0.56; 0.80), *p* = 0.88	0.56 (−0.23; 0.9), *p* = 0.14
APRI	0.33 (0.02; 0.59), *p* = 0.03	−0.15 (−0.45; 0.17), *p* = 0.35	0.26 (−0.54; 0.81), *p* = 0.52
FIB-4	0.25 (−0.07; 0.52), *p* = 0.12	−0.10 (−0.41; 0.22), *p* = 0.53	−0.20 (−0.79; 0.58), *p* = 0.62

Data are presented as Pearson coefficients, r (95% confidence interval, CI), *p*. * One patient was lost to follow-up after 4 weeks of treatment; transient elastography with LSM and CAP evaluation was performed in 8 patients, including 4 with baseline liver fibrosis >F1 and 4 patients without liver fibrosis (last patients included). ** One-year follow-up examination was performed in 3 patients with baseline liver fibrosis >F1 and in 5 patients without liver fibrosis. ALT—alanine aminotransferase; APRI—aspartate transaminase-to-platelet ratio index; AST—aspartate aminotransferase; BMI—body mass index; CAP—controlled attenuation parameter; EOT—end of treatment; FIB-4—fibrosis-4 index; LSM—liver stiffness measurement.

## Data Availability

The datasets used and analyzed during the current study are available from the corresponding author upon reasonable request.

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
