# Peer review of "The Influence of Treatment with Ledipasvir/Sofosbuvir on Growth Parameters in Children and Adolescents with Chronic Hepatitis C"

_viruses, 2022, doi:10.3390/v14030474_

Round 1

Reviewer 1 Report

This study mentions as increase in BMI as a marker of " impact of Sof/Ledipasvir" however this may not be accurate measure as BMI increase in age group 13/14 is expected to be higher than in age of 17over a period of 1 year

A subgroup analysis with age matched controls will help in better analysis of this data, this can be done by using the child development chart as reference and comparing this to the observed data.

Minor editing in English, Figure 1 mentions " Thinness" as  legend, better term to be used should "Lean" / "underweight"

Reviewer 2 Report

The authors studied 38 patients aged 10-17 years treated with LDV/SOF for CHC. A significant increase was observed in mean weight, height and BMI both 12 weeks and one year after the treatment compared to the baseline, whereas no differences were observed for BMI z scores and HA z scores. Baseline BMI z scores correlated with ALT level, LSM, APRI, and CAP. The information that LDV/SOF did not negatively influence the patients’ growth may be important, but is not so interesting.

Major comments

#1. The patients studied in this manuscript where the influences of LDV/SOF on growth were assessed are almost the same as those studied in their previous report where efficacy and side effects of LDV/SOF were studied [J Clin Med. 2021;10(18)]. They should clearly describe this fact.

#2. The number of the patients is too small for assessment of correlation of baseline BMI z scores with ALT level, LSM, APRI, and CAP. They should recruit a larger number of patients.

#3. One-year follow-up exam was performed in only 8 patients. This number is too small.

Minor comments

#1. Line 178. “Alanine aminotransferase” is 2nd or later appearance and should be written as ALT.

#2. Cirrhosis was diagnosed by LSM. The cut-off value should be indicated.

Round 2

Reviewer 2 Report

In the revised version of their manuscript, the authors have properly answered to the points raised by the reviewer.